# Maternal health care services utilization and associated factors among pregnant women in Kersa district, Jimma zone, Southwest Ethiopia

**Tsedach Alemu Abebe***, **Gurmesa Tura Debelew**

Department of Population and Family Health, Institute of Health, Jimma University, Jimma, Ethiopia

* tsedachalemu@yahoo.com

## Abstract

### Background

Ethiopia has a high rate of maternal mortality, documented at 412 deaths per 100,000 live births in 2016. One of the most important interventions to reduce maternal death from pregnancy-related problems is the use of maternal health care services. However, the utilization of these services continues to be low in rural areas of Ethiopia. Therefore, this study aimed to assess the utilization and its associated factors of maternal health care services among pregnant women in rural areas of Kersa district, Jimma zone, Southwest Ethiopia.

### Methods

A community-based cross-sectional study was conducted among 562 pregnant women in rural areas of Kersa District, Jimma Zone, Southwest Ethiopia, from October to December 2020. Cluster sampling was used to select study participants. The data was collected through face-to-face interviews using a pretested and structured questionnaire. The data was entered using Epi-data version 3.1 and SPSS version 21 was used for data cleaning, coding, labeling, and analysis. Bivariate and multi-variable logistic regression analyses were performed to determine the association between dependent and independent variables. The significance level for analyses was set at $p \le 0.05$.

### Results

Of the 562 women interviewed, 50.7% received at least one antenatal care (ANC) visit, more than two-thirds (70.5%) started ANC late, and only 2.8% completed ANC4 + visits. Nearly half (47.2%) of the women gave birth in a health facility. Women aged 20–29 years (AOR: 2.82; 95% CI: 1.27, 6.22) and 30–39 years (AOR: 1.86; 95% CI: 1.01, 3.42), women whose husbands were private employees (AOR: 2.23; 95% CI: 1.14, 4.38), women with good knowledge of pregnancy danger signs

**Data availability statement:** All relevant data are within the paper and its Supporting Information files.

**Funding:** The author(s) received no specific funding for this work.

**Competing interests:** The authors have declared that no competing interests exist.

**Abbreviations:** ACNM, American College of Nurse Midwives; ANC, Antenatal Care; ANC4+, Antenatal Care Visits Four and above; AOR, Adjusted Odds Ratio; BPCR, Birth Preparedness and Complication readiness plan; BP/CR, Birth Preparedness or Complication readiness plan; CI, Confidence Interval; COR, Crude Odds Ratio; EDHS, Ethiopian Demographic and Heath Survey; EMDHS, Ethiopian Mini Demographic and Heath Survey; ETB, Ethiopian Birr; FANC, Focused Antenatal Care; FIGO, International Federation of Gynecology and Obstetrics; HBLSS, Home Based Life Saving Skills; HAD, Health Development Army; HEW, Health Extension Workers; ICM, International Confederation of Midwives; JHPIEGO, Johns Hopkins Program for International Education in Gynecology and Obstetrics; MMR, Maternal Mortality Ratio; NGOs, None-Governmental Organization; PSU, Primary Sampling Unit; RCTS, Randomized Controlled Trial Studies; SBAs, Skilled Birth Attendants; SDGs, Sustainable Development Goals; SSA, Sub-Saharan Africa; SSU, Secondary Sampling Unit; TOT, Training of Trainers; WHO, World Health Organization.

(AOR: 1.99; 95% CI: 1.34, 2.94), good road conditions (AOR: 2.63; 95% CI: 1.67, 4.13), and low transportation costs (AOR: 3.16; 95% CI: 1.64, 6.08) were positively associated with antenatal care attendance.

Factors associated with institutional delivery service use were women aged 20–29 years (AOR: 3.47; 95% CI: 1.47, 8.17), private employee women (AOR: 5.46; 95% CI: 1.02, 29.17), women whose husbands were private employees (AOR: 2.38; 95% CI: 1.11, 5.11), households with a medium wealth index (AOR: 1.77; 95% CI: 1.05, 2.97), and rich households (AOR: 1.88; 95% CI: 1.09, 3.24), distance to health facility ≤ 1 hour (AOR: 4.95; 95% CI: 1.18, 20.76), and low transportation costs (AOR: 2.81; 95% CI: 1.51, 5.29).

## Conclusion

Maternal healthcare service utilization among pregnant women in the study area was lower than national targets and international recommendations. Therefore, increasing women's awareness of maternal healthcare services, addressing physical access to institutional delivery services, strengthening access to transportation such as making ambulance services more accessible, improving road conditions, and increasing household income would likely increase service utilization.

## Introduction

An estimated 287,000 women worldwide lost their lives to pregnancy or childbirth-related complications in 2020; with an average of 800 maternal deaths every day and approximately one every two minutes [1]. Ninety-nine percent of these deaths occur in developing nations, with Sub-Saharan Africa accounting for 70 percent of all deaths. Maternal mortality remains a persistent challenge [1].

In Ethiopia, the maternal mortality ratio (MMR) fell from a huge 676 deaths per 100,000 livebirths in 2011–412 deaths per 100,000 livebirths in 2016. However, the mortality rate is still far from reaching Goal 3 of the new Sustainable Development Goals (SDGs), which calls for a global mortality ratio of less than 70 deaths per 100,000 live births by 2030 [2]. For every woman who dies from pregnancy-related causes, many more suffer from morbidity, disabilities, and long-term ill health [3]. The death of a woman during pregnancy or childbirth is not only a health concern but also an issue of social injustice [4].

The majority of these maternal mortalities resulted from preventable causes. Evidence indicates that most obstetric complications are preventable and treatable with the use of maternal health care services [5]. Antenatal care (ANC) use during pregnancy and skilled attendance during delivery significantly contribute to the reduction of maternal mortality [6]. Approximately three-fourths of maternal near misses can be prevented with at least one ANC visit [7]. Around 16% to 33% of all maternal deaths may be avoided by skilled attendance at delivery [6].

Utilization of these services varied across different low- and middle-income countries. It was 7.7% in Benin, 31.8% in Bangladesh, 42.8% in Nepal, 43.3% in India,

and 95.8% in Armenia [8]. In Ethiopia, the 2019 Ethiopian Demographic Health Survey (EDHS) report indicated that 43% of women received four or more ANC visits, Only 28% of women had their first ANC visit during the first trimester, 50% of women were attended by a skilled birth attendant during delivery and 48% of women delivered in a health facility [9]. A study done in the Oromia region of Ethiopia indicated that institutional delivery was 74.7% [10]. In contrast, another study in this region showed that only 21.2% of deliveries were at health institutions [11]. According to Ethiopian Demographic Health Survey 2019 report urban Ethiopian women were more likely to use the services than rural; 72.1% of births to urban women were attended by skilled birth attendants and 70.4% of deliveries in health facilities. The percentages in rural areas were 42.5% and 40% respectively [9].

Several factors have been associated with service use, including socio-demographic factors – age, religion, residence, maternal education, husband's education, marital status, and employment status [12–16]. Other factors that influence include parity, birth order, household income, availability of service, poor quality of care, distance to the health facility, lack of family support, access to health information exposure, previous history of obstetric complications, cultural beliefs, the timing of first antenatal care and the number of antenatal care contact [17–19].

Despite a minor improvement in the utilization of antenatal and institutional delivery services in Ethiopia, rural women continue to use the services the least frequently as compared with urban women [9]. In addition, limited evidence exists on service utilization and its associated factors in rural areas of Ethiopia. Moreover, understanding the current level of utilization and identifying factors affecting are important to guide public health planners, policymakers, and implementers to plan and design appropriate intervention strategies. Therefore, this study aimed to assess maternal health care services utilization and its associated factors among pregnant women in rural areas of Kersa district, Jimma zone, Southwest Ethiopia.

## Methods

### Study design, area, and period

The study was conducted in rural areas of Kersa District, Jimma zone of the Oromia regional state in Southwest Ethiopia. Jimma zone is one of the 18 zones within the Oromia regional state of Ethiopia. It further divided into 18 rural districts known as *Woredas* and two town administrations. Serbo, the capital of Kersa district, is located 22 kms east of Jimma town and 324 kms from Addis Ababa, the capital city of Ethiopia. Kersa district consists of 31 Kebeles, which are the smallest formal administrative units. Based on the 2007 population and housing census, the total population of Jimma zone is 2.6 million, with women making up 49.9% of the population. Eighty-nine percent of the population are rural residents and farmers in occupation. A community-based cross-sectional study was conducted as part of baseline assessment of a big follow up study. The study was conducted from October 23 to December 17, 2020.

### Study population

The study participants for this study were pregnant women with 12–24 weeks of gestational age, previously gave birth at least once and with most recent births preceding the survey.

### Sample size calculation

This is the baseline assessment of a big follow up study. Therefore, the sample size was determined using a clustered randomized trial formula for a binary outcome and based on the following assumptions:-The null hypothesis is $H_O$: P1-P2 = 0 and the alternative hypothesis is $H_1$: P1-P2 ≠ 0, Primary outcome = The status of institutional delivery service utilization in Oromia region was 41% [9], two-sided α = 0.05 significance level, Z alpha value = 1.96, Z beta value = 0.84, power of 80% to detect at least 15% difference between the two groups after the intervention. The assumed sampling ratio between the control and intervention group is 1:1. Using design effect 2 and assuming a

10% non-response rate, the final sample size was estimated to be 686 respondents. The design effect was determined using an ICC of 0.03; which was taken from a previous similar study [20], the average cluster size of 34, and. assuming equal cluster sizes.

## Sampling procedure

A clustered sampling technique was used to identify and enroll participants in the study. Initially, a total of 31 kebeles (clusters) in the district were listed, from which 20 kebeles were randomly selected. In the second stage, home–to–home enumeration (census) was conducted by female enumerators who had completed grade 10th, using a pregnancy screening checklist (Adapted from Stanback et al, 1999). Finally, multigravida pregnant women with a gestational age of 12–24 weeks, who previously gave birth at least once and with most recent births preceding the survey were included in the study. Written informed consent was obtained from each woman before their inclusion in the study. The data collectors explained all procedures using an information sheet. The women were allowed to ask questions and relevant information was provided accordingly. Women who were willing to consent were either signed or put on their fingerprints, depending on their literacy status.

## Operational definitions

### Study variables. The dependent variables

**Maternal healthcare services**: maternal healthcare services include antenatal care and institutional delivery services.

**Antenatal care (ANC):** is the healthcare provided to pregnant women by skilled healthcare professionals to ensure optimal health for both mother and baby during pregnancy [21].

**Institutional delivery**: institutional delivery means giving birth to a child in a medical institution under the overall supervision of trained and competent health workers, where amenities are available to manage the situation and save the lives of mother and child.

**Independent variables**

**Socio-demographic variables:** age, religion, ethnicity, marital status, educational status of husband, occupation status of husband, education status of wife, occupation of wife and family size.

**The wealth index** was used in a measure of the socio-economic status of mothers. It was calculated from ownership of various household resources including the type of flooring, walls and roof materials used in the house, the number of rooms in the household, the presence of separate kitchen rooms, the source of drinking water, water source for cooking, toilet facilities, type of fuel used for cooking, ownership of agricultural land and livestock such as cattle, sheep, goats, milk cows, oxen, horses, donkey, chicken and small equipment, including radio, television, electricity, bicycle, motorcycle, and car, Six principal components with eigenvalues greater than one were summed to obtain wealth index values after principal component analysis (PCA) was run. The resulting index was then divided into three categories representing the first quantile (poor), second quantile (medium), and third quantile (rich).

**Health facility-related factors**: distance to the nearest healthcare facility, health service quality, accessibility of adequate transportation, cost of transport, and condition of roads.

**Low transportation costs:** it was measured from the report of women, that if women reported the transportation cost was low.

**Good road condition:** this was measured from the report of women, that if women reported the road condition was good.

**Perceived quality of care:** is defined as women's opinion about the overall quality or image of the services.

**Distance to the nearest health facility** this was measured from the report of women on the walking hours to the health facilities. This was coded 1 if women reported distance in walking hours ≤ 1 hour and coded as 0 if women reported >1 hour.

 

**Maternal obstetric factors:** parity, gravidity and history of previous obstetric complications.

**Knowledge of danger signs:** respondents, whose knowledge of key danger signs related to pregnancy, childbirth and the postpartum period above the mean score were categorized as having good knowledge; whereas respondents below the mean score were categorized as having poor knowledge about danger signs.

**Women's autonomy:** head of household, financial independence, Mobility and women's decision-making power were asked.

### Data collection procedures and quality control

A structured questionnaire was used to collect the data which was developed by reviewing different literature. The data were collected through face-to-face interviews. The questionnaire was initially prepared in English and translated to the local language (Afan Oromo) and back-translated to English by language experts to see the consistency. The questionnaire was pretested on 5% of the sample size outside the study setting. Twenty (one per kebele) female data collectors who completed grade 10th were collected the data. Similarly, two bachelor's degree holders in health discipline supervise the data collection. Three days of intensive training were given to data collectors and supervisors regarding the objective and method of data collection and discussed the presence of unclear questions in the questionnaire.

In addition, the day-to-day activities of the data collection was closely supervised by the investigators. The data entry template was designed in Epi-data version 3.1 to control error entries.

### Data processing and analysis

Data were checked, cleaned, coded, and entered into Epi Data version 3.1 and analyzed using SPSS version 21. Both descriptive statistics and multivariable analytical models were used to summarize the data and identify the association of independent variables with the outcome of the study. Adjusted odds ratios (AORs) with their corresponding 95% CIs were used to assess the strength of associations at a *p-value* ≤ 0.05 cut–off point for statistical significance.

### Ethical approval and consent to participate

Ethical clearances were obtained from the institutional review board of Jimma University institute of health sciences, with Ethical approval number of Ref No. IHRPGD/854/20. The purpose of the trial was explained in a formal letter and verbally to district administrative and health officials. Approval to include the selected kebeles in the trial was sought from kebele authorities.

Written informed consent was obtained from each respondent before actual data collection after reading the information sheet in local language. Women who were willing to consent were either signed or put their finger-prints according to their literacy status. Issues of confidentiality were maintained by removing any identifiers from the questionnaire and the right to refuse or withdraw at any time was respected. This is a baseline assessment for health intervention the study would not adversely affect the study participants/institutions; rather, it is expected to improve the health and well-being of mothers.

## Results

### Socio-demographic characteristics of respondents

Of the 686 pregnant women recruited, 562 were successfully completed the interview, making a response rate of 82%. The majority of the participants 336(59.8%) were in the age group of 20–29 years, 553(98.9%) were Muslims, Oromo ethnic group 560(99.6%), housewives 540(96.1%), 267(47.5%) no formal education, 283(50.4%) family size 1–3 and 189(33.6%) of respondents' wealth index were in the third quantile (Table 1).

### Maternal health care services utilization of participants

**Antenatal care services utilization.** Out of 562 participants, 285(50.7%) attended at least one antenatal care visit in their previous pregnancy, 84(29.5%) started first ANC visit in the first trimester of pregnancy. only 16(2.8%) completed

**Table 1.  Socio-demographic characteristics of pregnant women in Kersa district, Jimma zone, Southwest Ethiopia, 2020 (n = 562).**

| Variables | frequency | percentage |
|---|---|---|
| Age in years | | |
| 20-29 | 336 | 59.8 |
| 30-39 | 219 | 39 |
| 40-49 | 7 | 1.2 |
| Religion | | |
| Muslim | 553 | 98.9 |
| Christian | 6 | 1.1 |
| Ethnicity | | |
| Oromo | 560 | 99.6 |
| Amhara | 2 | 1.1 |
| Women education level | | |
| No formal education | 267 | 47.5 |
| Formal education | 295 | 52.5 |
| Women occupation | | |
| Housewife | 540 | 96.1 |
| Private employee | 22 | 3.5 |
| Husband education level | | |
| No formal education | 245 | 43.6 |
| Formal education | 317 | 56.4 |
| Husband occupation | | |
| Farmer | 487 | 86.7 |
| Private employee | 75 | 15.3 |
| Family size | | |
| 1-3 | 214 | 38.1 |
| 4-6 | 283 | 50.4 |
| ≥7 | 85 | 11.6 |
| Wealth index | | |
| First quantile (poor) | 187 | 33.3 |
| Second quantile (middle) | 186 | 33.1 |
| Third quantile (rich) | 189 | 33.6 |

the recommended at least four ANC visits. Majority 187(33.3%) of women ANC attendant health worker was nurses. Two hundred sixty five (93%) attended their ANC in government health center. One hundred seventy six (61.8%) were accompanied by their husbands. Only 71(24.9%) respondents were health worker advised on danger signs at ANC attendance time (Table 2).

## Institutional delivery services utilization

Among 562 respondents, 265(47.2%) gave their last birth in health facility. About 284(50.5%) of respondents planned their last birth place. Majority 102(38.5%) respondents used ambulance to went to health facility, 222(83.8%) of respondents accompanied by their husbands for health facility delivery. Only 45(8.0%) decided jointly to place of delivery. 182(68.7%) participants birth attendant health worker were midwives (Table 3).

**Table 2. Antenatal care services utilization among pregnant women in Kersa district, Jimma zone, Southwest Ethiopia, 2020(n = 562).**

| Variables | frequency | Percentage |
|---|---|---|
| Gravidity | | |
| 2-4 pregnancies | 345 | 61.4 |
| ≥5 pregnancies | 217 | 38.6 |
| Parity(n = 562) | | |
| 1 birth | 101 | 18 |
| 2-4 births | 341 | 60.7 |
| ≥5 births | 120 | 21.3 |
| ANC attended in last pregnancy (at least once) (n = 562) | | |
| Yes | 285 | 50.7 |
| No | 277 | 49.3 |
| Number of ANC visit (n = 562) | | |
| 0 | 277 | 49.3 |
| 1 visit | 50 | 8.9 |
| 2 visits | 182 | 32.4 |
| 3 visits | 37 | 6.6 |
| ≥ 4 visits | 16 | 2.8 |
| Time of first ANC visit (n = 562) | | |
| First trimester | 85 | 15.1 |
| Second trimester | 193 | 34.3 |
| Third trimester | 8 | 1.4 |
| Do not know | 277 | 49.3 |
| ANC attendant health professional (n = 562) | | |
| Doctor | 6 | 1.1 |
| Midwife | 89 | 15.8 |
| Nurse | 187 | 33.3 |
| Health officer | 3 | 0.5 |
| Do not know | 277 | 49.3 |
| Place of ANC attendance (n = 285) | | |
| Government hospital | 20 | 0.7 |
| Government health center | 265 | 93.0 |
| Husband accompanied(n = 285) | | |
| Yes | 176 | 61.8 |
| No | 109 | 38.2 |
| Health worker advised on danger signs | | |
| Yes | 71 | 24.9 |
| No | 214 | 75.1 |
| Advised where to go if danger signs happened (n = 285) | | |
| Yes | 107 | 37.5 |
| No | 147 | 41.6 |
| Advised on place of delivery (n = 285) | | |
| Yes | 138 | 48.4 |
| No | 147 | 41.6 |
| Advised on identifying a skill provider (n = 285) | | |
| Yes | 7 | 2.5 |
| No | 278 | 98.5 |

*(Continued)*

**Table 2.** (Continued)

| Variables | frequency | Percentage |
|---|---|---|
| Experienced pregnancy complications (n = 562) | | |
| Yes | 46 | 8.2 |
| No | 516 | 91.8 |
| Experienced severe vaginal bleeding during pregnancy (n = 562) | | |
| Yes | 25 | 4.4 |
| No | 537 | 95.6 |

### Factors affecting utilization of maternal healthcare services

Occupation of wife, family size, wealth index, health service quality, distance to health facility and adequate transport available did not show a significant association on ANC services utilization of women at multivariable logistic regression level analysis. On the other hand, the age of women, occupation of husband, knowledge of danger signs during pregnancy, road conditions, and cost of transport showed significant association with ANC services utilization of women at multivariable logistic regression level analysis. There is a significant association between the age of woman and ANC attendance, where women with ages of 20–29 years (AOR: 2.82; 95% CI: 1.27, 6.22; $P$ = 0.01) and 30–39 years (AOR: 1.86; CI: 1.01, 3.42; $P$ = 0.045) more likely attended ANC compared to mothers with age of 40–49 years.

Women with private employee husbands were more than two times (AOR: 2.23; CI: 1.14, 4.38; $P$ = 0.02) more likely utilize ANC services than women whose husband's occupation were farmers. Women having good knowledge on danger signs during pregnancy nearly two times (AOR: 1.99; 1.34, 2.94; $P$ = 0.001) more likely attended ANC compared to mothers having poor knowledge of danger signs during pregnancy. Good road condition (AOR: 2.63; 1.67, 4.13; $P$ = 0.0001), easy to pay on transportation cost (AOR: 3.16; CI: 1.64, 6.08; $P$ = 0.001) were positively associated with ANC services utilization (Table 4).

Factors like educational level of wife and husband, family size, knowledge of pregnancy danger signs and women decision making power on use of health care services did not significantly associate with institutional delivery service utilization at multivariable logistic regression level analysis. Factors like age of women, occupation of wife and occupation of husband, wealth index, distance to health facility, road condition and cost of transportation were significantly associated with institutional delivery service utilization at multivariable logistic regression level analysis. Women with ages 20–29 years were four times (AOR: 4.00; CI: 1.78, 8.91; $P$ = 0.001) more likely to give birth at health institutions compared to 40–49 years old women. Private employee mothers were more than five times (AOR: 5.46; CI: 1.10, 26.09; $P$ = 0.038) more likely to give birth at a health institution compared to mothers with the occupation of housewives. Mothers whose husbands were private employees were more than three times (AOR: 3.07; CI: 1.51, 6.24; $P$ = 0.002) more likely to utilize institutional delivery services compared to mothers whose husbands' occupations were farmers. Women in wealth index of the second quantile (AOR; 1.74; CI: 1.07, 2.81; $P$ = 0.025) and third quantile (AOR: 1.89; CI: 1.14, 3.15; $P$ = 0.014) were nearly two times more likely give birth at health institution compared to women whose wealth index is in the first quantile. Good road condition (AOR: 2.63; CI: 1.67, 4.13; $P$ = 0.0001), less than or equal to one hour on distance to health facility (AOR: 4.95; CI: 1.18, 20.76; $P$ = 0.029) and easy to pay on transportation cost (AOR: 2.81; CI: 1.51, 5.29; $P$ = 0.001) were positively associated with give birth at health institution (Table 5).

## Discussion

The use of maternal healthcare services plays a pivotal role in decreasing maternal mortality rates. Evidence has indicated that receiving care from a skilled attendant during pregnancy and childbirth is strongly linked to improved maternal survival rates [5–7]. The current study tried to assess the status of maternal healthcare services utilization and its associated factors among pregnant women in Kersa district, Jimma Zone, Southwest Ethiopia.

This study found that just more than half (50.7%) of women attended antenatal care at least once during their previous pregnancy, 29.5% initiated antenatal care at or before the 12th weeks of gestation, only 2.8% of women had received four or more ANC visits and 47.2% used institutional delivery care. age of women, husband's occupation, knowledge of danger signs during pregnancy, road conditions, and transportation cost were pertinent predictors of ANC utilization. Age and occupation of women, occupation of husband, household wealth index, distance to health facility, road conditions, and transportation cost were significant predictors of institutional delivery service utilization.

In this study, 50.7% of women attended antenatal care at least once during their last pregnancy. This finding was lower than studies done in Kenya 62.7% [22] and Nepal 76% [23] and also lower than local studies done in Ethiopia, the national Mini EDHS 2019 report of 74% [9], Jimma zone southwest Ethiopia at 93.3% [24]. However, the result was higher Somali pastoral communities of Eastern Ethiopia 27% [25]. The observed difference might be due to a difference in sample size, the study setting, the study period, and the accessibility of health services. Besides this, variations in socio-cultural, women's health-seeking behavior and awareness differences might contribute to this difference.

Concerning the timing of the first ANC visit, this study showed that 29.5% of women initiate antenatal care at or before the 12th week of gestation. This finding is consistent with the national Mini EDHS 2019 report of 28% [9]. However, it is lower than a study done in Southern Ethiopia 38.96% [26] and Addis Ababa Ethiopia 50.3% [27]. This might be because of the lack of adequate information on the timing of first antenatal care. This study revealed that nearly half of 47.2% of women delivered their last child in health facilities. This finding was consistent with the 2019 Ethiopian Demographic Health Survey (EDHS) report which was 50% [9]. The value was lower than studies done in Jimma Zone Ethiopia 77.4% [24], Bahir Dar City Ethiopia 78.8% [28], Central Gondar Zone 58.17% and Ghana 77.89% [29]. The possible explanation might be due to differences in awareness creation regarding institutional delivery services, access to health services, sociocultural factors, and level of urbanization across studies. On the other hand, the finding was higher than the study findings of Sidama Zone, Southeast Ethiopia 26.4% [30], and Sekela District, Northwest Ethiopia 12.1% [31]. The possible reason for the discrepancy is that this study was conducted after the Ethiopian government started free delivery services at all levels of health facilities and another reason that might have contributed to this difference is the study period.

The study found that Younger aged women were nearly three times more likely use ANC services than elders in the current study and other studies reported similar finding [32,33]. This could be explained by the fact that young women at their first pregnancy are more careful about their pregnancy and therefore require institutional care more than older women. In contrast; older women a more self-confidence and gain experience from earlier pregnancies; so, they are less likely to utilize antenatal care services.

Husbands' occupation was significantly associated with antenatal care services utilization. Women with private employee husbands were more than two times more likely to utilize antenatal care services than farmer husbands. This finding is consistent with a study done in the Tigray region of Ethiopia [34]. One possible explanation is that men engaged in agricultural labor work extended hours and, thus, are, unable to transport or accompany their spouses to the clinics.

Women having good knowledge of danger signs during pregnancy were nearly two times more likely to utilize antenatal care services compared to women having poor knowledge of danger signs during pregnancy. This finding is consistent with other study findings [34]. This might be because Knowledge is an important factor that affects attitude, intention, and behavior. Women who have sufficient knowledge about pregnancy danger signs might have perceived the service benefits of a health institution.

The road condition is significantly associated with ANC services utilization. Women who reported good road conditions were more than two times more likely to use antenatal care services compared to those women who reported poor road conditions. This study finding is similar to other study findings where the lack of proper roads was a reason for not using ANC services [19].

Transportation cost is one of the factors affecting antenatal care services utilization. This study indicated that women whose ability to pay transportation costs were more than three times more likely to utilize antenatal care services

**Table 3. Institutional delivery services utilization among pregnant women in Kersa district, Jimma zone, Southwest, Ethiopia, 2020(n=562).**

| Variables | Frequency | Percentage |
|---|---|---|
| Facility delivery (n=562) | | |
| Yes | 265 | 47.2 |
| No | 297 | 52.8 |
| Place of delivery (n=562) | | |
| Government hospital | 42 | 7.5 |
| Government health center | 223 | 39.7 |
| Home | 297 | 52.8 |
| Planned place of delivery (=562) | 284 | 50.5 |
| Yes | 278 | 49.5 |
| No | | |
| Decision maker for place of delivery (n=562) | | |
| No one | 235 | 41.8 |
| Wife | 70 | 12.5 |
| Husband | 212 | 37.7 |
| Jointly | 45 | 8.0 |
| Transport used to go to health facility (n=265) | | |
| Ambulance | 102 | 38.5 |
| Private car | 16 | 6.0 |
| Bajaj | 51 | 19.2 |
| Locally made streachers | 18 | 6.8 |
| On foot | 78 | 29.4 |
| Accompanier to facility delivery (n=265) | | |
| Husband | 222 | 83.8 |
| Respondent mother | 31 | 11.7 |
| Mother in law | 12 | 4.5 |
| Time taken to get health services (265) | | |
| Immediately | 207 | 78.1 |
| ≤ 1 hour | 39 | 14.7 |
| >1 hour | 19 | 7.2 |
| Birth attendant health worker (265) | | |
| Doctor | 23 | 8.7 |
| Midwife | 182 | 68.7 |
| Nurse | 57 | 21.5 |
| Health officer | 3 | 1.1 |
| Child born by cesarean section (265) | | |
| Yes | 5 | 1.9 |
| No | 260 | 98.1 |
| Children born by forceps/vacuum (n=265) | | |
| Yes | 22 | 8.3 |
| No | 243 | 91.7 |
| Experienced birth complications (n=265) | | |
| Yes | 17 | 6.4 |
| No | 248 | 93.6 |
| Type of health complications experienced (n=17) | | |
| Severe bleeding | 12 | 70.6 |
| Severe headache | 3 | 17.6 |

*(Continued)*

**Table 3.** (Continued)

| Variables | Frequency | Percentage |
|---|---|---|
| High fever | 1 | 5.9 |
| Loss of consciousness | 1 | 5.9 |

utilization compared to those women with difficult-to-pay transportation costs. This finding is consistent with other study findings which reported that high transportation costs hindered them from accessing maternal healthcare services [19].

The study revealed that women in the age group of 20–29 years were four times more likely to use institutional delivery compared with women in the age group of 30–39 and 40–49 years. The finding was in line with other studies' findings [35,36]. This might be because younger women are more likely to be educated and they may have a better opportunity to access information as compared to older women. Moreover, home delivery may not be considered risky for most of the older women who have previously experienced birth at home.

Women's and husband's occupations were significantly associated with institutional delivery service use. This study revealed that private employee women were more than five times more likely to use institutional delivery services compared with housewives. This finding was in line with the findings of other studies, where the results of these studies indicated that women in occupations other than housewives (merchants, employees, farmers, and private businesses) were more likely to use institutional delivery compared to housewives [35,37]. This could be the fact that these women had better educational status than housewives in most of the cases and were economically decided independently. Moreover, having own income helps to empower women to decide on their health. Women whose husbands occupations private employees were more than three times more likely to give birth in a health institution than farmer husbands. This finding was consistent with other study findings [38]. This could be the fact that Women from families in good economic condition are more likely to give birth in health facilities.

In this study, another important significant predictor of institutional delivery service use was the wealth index. Women who were from households with middle and rich wealth quantiles were nearly two times more likely to use institutional delivery than women who were from poor households. It was consistent with other previous studies' findings [29,39–42]. The possible explanation could be household wealth has the potential to influence women's decisions regarding the place of delivery, access to healthcare services, transportation, and additional costs. Women who can afford to pay for such costs are more likely to visit health facilities than poor women.

The study found that distance to the health facility was significantly associated with institutional delivery service utilization. In this study, women who resided a walking distance to reach the nearest health facility less or equal to one hour were nearly seven times more likely to use institutional delivery services compared to women who resided greater than one hour. This result was in line with other studies findings [43–46]. The possible reason behind this fact could be the distance to healthcare facilities can impact the costs of accessing care due to transportation expenses and affect a woman's ability to easily reach the health facility.

The study showed that transportation costs and road conditions were significantly associated with institutional delivery service utilization. Women whose ability to pay transportation costs were nearly four times more likely to utilize health services compared with those women with difficulty paying transportation costs. The result was in line with previously done other study findings [47,48]. The study found that women with good road conditions were more than one time more likely to use institutional delivery services than poor road conditions. The finding was supported by another previous study [48].

## Conclusions

In conclusion, maternal health care services utilization was lower than national targets and international recommendations. The majority of the women started ANC lately and had not completed the recommended numbers of ANC visits.

**Table 4. Factors affecting antenatal care services utilization among pregnant women in Kersa district, Jima zone, Southwest Ethiopia, 2020.**

| Variables | ANC service Utilization Yes No | Crud OR 95% CI | Adjusted OR 95% CI | P values (adjusted) |
|---|---|---|---|---|
| Age | | | | |
| 20-29 | 171(50.9) 165(49.2) | 2.5(1.16,4.75) | 2.82(1.27,6.22) | 0.01 |
| 30-39 | 112(51.1) 107(48.9) | 1.38(0.81,2.35) | 1.86(1.01, 3.42) | 0.045 |
| 40-49 | 2(28.6) 5(71.4) | 1 | 1 | |
| Occupationofwife | | | | |
| House wife | 268(49.6) 272(50.4) | 1 | 1 | |
| Privateemployee | 17(77.3) 5(22.7) | 3.45(1.26, 9.49) | 1.57(0.48, 5.16) | 0.461 |
| Occupation of husband | | | | |
| Farmer | 227(46.6) 260(53.4) | 1 | 1 | |
| Private employee | 58(77.3) 17(22.7) | 3.91(2.21, 6.9) | 2.23(1.14, 4.38) | 0.02 |
| Family size | | | | |
| 1-3 | 127(59.3) 87(40.7) | 2.85(1.6, 5.1) | 1.40(0.71, 2.77) | 0.329 |
| 4-6 | 136(48.1) 147(51.9) | 1.81(1.03, 3.18) | 1.0(0.53, 1.89) | 0.993 |
| ≥ 7 | 22(33.8) 43(66.2) | 1 | 1 | |
| Wealth index | | | | |
| First quantile | 79(42.2) 108(57.8) | 1 | 1 | |
| Second quantile | 101(54.3) 85(45.7) | 1.62(1.08, 2.45) | 1.54(0.96, 2.47) | 0.073 |
| Third quantile | 105(55.6) 84(44.4) | 1.71(1.14, 2.57) | 1.30(0.79, 2.12) | 0.302 |
| Knowledge of danger signs during pregnancy | | | | |
| Poor knowledge | 110(40.7) 160(59.3) | 1 | 1 | |
| Good knowledge | 175(59.9) 117(40.1) | 2.18(1.55, 3.05) | 1.99(1.34, 2.94) | 0.001 |
| Health service quality | | | | |
| Excellent | 100(74.6) 34(25.4) | 2.71(1.13, 6.52) | 2.03(0.73, 5.60) | 0.173 |
| Good | 172(42.7) 231(57.3) | 0.69(0.31, 1.54) | 0.88(0.34, 2.30) | 0.799 |
| Poor | 13(52) 12(48) | 1 | 1 | |
| Distance to health facility | | | | |
| ≤ 1 hour | 15(75) 5(25) | 3.02(1.08, 8.43) | 2.52(0.80, 7.94) | 0.115 |
| >1 hour | 270(49.8) 272(50.2) | 1 | 1 | |
| Road condition | | | | |
| Good | 171(68.1) 80(31.9) | 3.69(2.6, 5.25) | 2.63(1.67, 4.13) | 0.000 |
| Poor | 114(36.7) 197(63.3) | 1 | 1 | 1 |
| Adequate transport available | | | | |
| Yes | 187(61.5) 117(38.5) | 2.61(1.86, 3.67) | 0.83(0.51, 1.32) | 0.423 |
| No | 98(38.0) 160(62.0) | 1 | 1 | |
| Transportation cost | | | | |
| Low | 74(75.5) 24(24.5) | 3.70(2.25, 6.07) | 3.16(1.64, 6.08) | 0.001 |
| High | 211(45.5) 253(54.5) | 1 | 1 | |

More than half of the mothers delivered at home without the support of skilled attendants. The age of women, husband's occupation, knowledge of danger signs during pregnancy, condition of roads and transportation costs were factors significantly affecting antenatal care services utilization. Age of women, occupation of women, occupation of husband, household wealth index, distance to health facility, road conditions, and transportation cost were significantly associated with institutional delivery service utilization.

**Table 5. Factors affecting institutional delivery services utilization among pregnant women in kersa district, Jimma zone, Southwest Ethiopia, 2020(n = 562).**

| Variables | Institutional delivery Service utilization Yes No | Crude OR 95% CI | Adjusted OR 95% CI | *P value* (Adjusted) |
|---|---|---|---|---|
| Age | | | | |
| 20-29 | 162(48.2) 174(51.8) | 3.26(1.66, 6.39) | 4.00(1.79, 8.91) | 0.001 |
| 30-39 | 100(56.7) 119(54.3) | 1.28(0.75, 2.19) | 1.76(0.93, 3.33) | 0.081 |
| 40-49 | 3(42.9) 4(51.1) | 1 | 1 | |
| Education level of wife | | | | |
| No formal education | 102(38.2) 165(61.8) | 1 | 1 | |
| Formal education | 163(55.3) 132(44.7) | 2.0(1.43, 2.80) | 1.54(0.92, 2.58) | 0.102 |
| Education level of husband | | | | |
| No formal education | 103(42.0) 142(58.0) | 1 | 1 | |
| Formal education | 162(51.1) 155(48.9) | 1.44(1.03, 2.03) | 0.81(0.49, 1.36) | |
| Occupation of wife | | | | |
| House wife | 245(45.4) 295(54.6) | 1 | 1 | |
| Private employee | 20(90.9) 2(9.1) | 0.08(0.02, 0.36) | 5.36(1.10, 26.09) | 0.038 |
| Occupation of husband | | | | |
| Farmer | 204(41.9) 283(58.1) | 1 | 1 | |
| Private employee | 61(81.3) 14(18.7) | 6.04(3.29, 11.10) | 3.07(1.51, 6.24) | 0.002 |
| Family size | | | | |
| 1-3 | 126(58.9) 88(41.1) | 3.5(1.9, 6.3) | 1.69(0.82, 3.51) | 0.152 |
| 4-6 | 120(42.4) 163(57.6) | 1.8(1.0, 3.2) | 1.05(0.53, 2.09) | 0.886 |
| >=7 | 19(29.2) 46(70.8) | 1 | 1 | |
| Wealth index | | | | |
| First quantile (poor) | 70(37.4) 117(62.6) | 1 | 1 | |
| Second quantile(medium) | 90(48.4) 96(51.6) | 1.57(1.04, 2.37) | 1.74(1.07, 2.81) | 0.025 |
| Third quantile (rich) | 105(55.6) 84(44.4) | 2.09(1.38, 3.16) | 1.89(1.14, 3.15) | 0.014 |
| Knowledge of pregnancy danger signs | | | | |
| Poor knowledge | 119(44.1) 151(55.9) | 1 | 1 | |
| Good knowledge | 156(50.0) 156(50.0) | 1.27(0.91, 1.77) | 1.31(0.68, 2.53) | 0.420 |
| Women decision making power on use of health services | | | | |
| Yes | 46(55.4) 37(44.6) | 1.48(0.92, 2.36) | 0.98(0.54, 1.75) | 0.935 |
| No | 219(45.7) 260(54.3) | 1 | 1 | |
| Distance to health facility | | | | |
| <=1 hour | 17(85.0) 3(15.0) | 8.26(2.25, 23.2) | 6.88(1.80, 26.26) | 0.005 |
| >1hour | 248(45.8) 294(54.2) | 1 | 1 | |
| Road condition | | | | |
| Good | 149(59.4) 102(40.6) | 2.5(1.8, 3.5) | 1.66(1.09, 2.52) | 0.018 |
| Poor | 116(37.3) 195(62.7) | 1 | 1 | |
| Transportation cost | | | | |
| Low | 76(77.6) 22(22.4) | 5.03(3.02, 8.37) | 3.92(2.24, 6.88) | 0.0001 |
| High | 189(40.7) 275(59.3) | 1 | 1 | |

## Recommendations

Therefore, increasing women's awareness of maternal healthcare services, addressing physical access to institutional delivery services, strengthening access to transportation such as making ambulance services more accessible, improving road conditions, and increasing household income would likely increase service utilization. Finally, this study recommends further (qualitative) research to explore barriers to maternal healthcare services utilization.

## Strengths and limitations of the study

The strength of this study is that being community-based, it could reflect the actual experience of women during the study period. The limitations of this study are the methodological nature of the cross-sectional study design limited the causal inference of the study variables. Another limitation is the study might be prone to recall bias because the information was collected by the study participants' self-report.

## Supporting information

**S1 Data. Raw survey data used for statistical analysis in this study.**
(XLSX)

## Acknowledgments

I would like to acknowledge Jimma University and the Department of Population and Family Health. My gratitude goes to the study participants and research assistants. I wish to acknowledge Kersa District Health Office staff.

## Author contributions

**Conceptualization:** Tsedach Alemu Abebe, Gurmesa Tura Debelew.

**Data curation:** Tsedach Alemu Abebe, Gurmesa Tura Debelew.

**Formal analysis:** Tsedach Alemu Abebe, Gurmesa Tura Debelew.

**Funding acquisition:** Tsedach Alemu Abebe.

**Investigation:** Tsedach Alemu Abebe.

**Methodology:** Tsedach Alemu Abebe, Gurmesa Tura Debelew.

**Project administration:** Tsedach Alemu Abebe.

**Resources:** Tsedach Alemu Abebe.

**Software:** Tsedach Alemu Abebe.

**Supervision:** Tsedach Alemu Abebe, Gurmesa Tura Debelew.

**Validation:** Tsedach Alemu Abebe.

**Visualization:** Tsedach Alemu Abebe.

**Writing – original draft:** Tsedach Alemu Abebe.

**Writing – review & editing:** Tsedach Alemu Abebe.

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
