## [Decision Letter · Decision Letter 0]

3 Dec 2024

PONE-D-24-46689Maternal Health Care Services Utilization Among Pregnant Women in Kersa District, Jimma Zone, Southwest, Ethiopia.PLOS ONE

Dear Dr. Abebe,

Thank you for submitting your manuscript to PLOS ONE. After careful consideration, we feel that it has merit but does not fully meet PLOS ONE’s publication criteria as it currently stands. Therefore, we invite you to submit a revised version of the manuscript that addresses the points raised during the review process.

We look forward to receiving your revised manuscript.

Kind regards,

Kahsu Gebrekidan, Ph.D.

Academic Editor

PLOS ONE

Reviewers' comments:

Reviewer's Responses to Questions

**Comments to the Author**

1. Is the manuscript technically sound, and do the data support the conclusions?

Reviewer #1: Partly

Reviewer #2: No

2. Has the statistical analysis been performed appropriately and rigorously? 

Reviewer #1: Yes

Reviewer #2: Yes

3. Have the authors made all data underlying the findings in their manuscript fully available?

Reviewer #1: Yes

Reviewer #2: Yes

4. Is the manuscript presented in an intelligible fashion and written in standard English?

Reviewer #1: No

Reviewer #2: No

5. Review Comments to the Author

Reviewer #1: General comments

Authors should review the manuscript and correct some grammatical errors and the manuscript needs basic editing throughout?

Topic: If the study includes a specific period (e.g., last one year), it would help make the topic more precise and indicate a timeframe for analysis.

1. Abstract

A. Background

The statement notes that maternal mortality is a global concern and provides a general statistic (99% of preventable deaths in developing countries). It could be more precise by providing either recent data specific to Ethiopia or clarifying why Ethiopia’s situation is particularly concerning? The phrase “maternal health care service utilization remains suboptimal” is somewhat vague. Specify in what way utilization is suboptimal?

B. Methods: The sample of 562 pregnant women is specified, but the sampling method is not mentioned?

The statement "variables with a p-value of p≤0.05 were considered statistically significant" is clear, but it could be condensed by eliminating redundancy?

C. Results: The findings could be organized by outcome (e.g., ANC visits and institutional delivery) for clarity. Separating them into distinct sections may enhance readability.

Inconsistent terms (like "mothers" and "women") can disrupt the flow. Choose one term and stick to it for clarity?

D. Conclusion: you conclude that “Maternal health care service utilization among pregnant women in the study area was found to be low” but what was your standard reference to say low?

2. Introduction: The study could be strengthened by emphasizing the gap in local data and the importance of region-specific insights?

The disparities in ANC and institutional delivery rates between urban and rural women are repeated in multiple sentences. Condense this information for clarity?

The benefits of maternal health services are stated multiple times in slightly different ways, which can be repetitive. Reorganization these points can enhance readability?

The trend in Ethiopia’s maternal mortality ratio is mentioned, but the comparison with SDG goals could be clarified more?

3. Methods: The description of the intervention could be clearer. It is also helpful to specify what the "standard care" entails for the control group? Clarify whether informed consent was verbal or written? You describe as your study participants for this study were pregnant women with 12 to 24 weeks of gestational age but why? Needs to operationalize the following (low transportation costs, good road conditions

4. Result: The response rate is very low (82%). Did you use different methods to increase response rate? Make the tables standardized? Clarify the type of health workers seen and the location of ANC visits. For example, "nurses" might include various roles in different settings?

5. Discussions

Unclear and vast discussions. The discussion tries to compare and contrast the findings from other research findings which are not similar with the current study (in the context of study population, setting…

Justification of the discrepancy is not as such strong? Please make it clear and precise

6. Recommendations

Recommendation should be based on your findings?

Your limitations and strengths are not rational?

Reviewer #2: My Comments to authors

Thank you very much for giving me the opportunity to review this paper. In general, it requires serious amendments. Here below are my comments to improve the paper:

Abstract

1. Key words should be revised-based MeSH terms. Are Kersa District and Jimma Zone important as key terms?

2. The result in the abstract needs some modification in such a way that it is clear for all readers.

3. Conclusion and recommendation not support the finding of the article.

4. As written in the submission guidelines of PLOS ONE, you have to include the line number to make easy reviewing.

Introduction

1. first two sentences need citation

2. You have started the sentence like ‘99% of mortality...’ Please edit for the grammar and spacing

3. Please be consistent in the reference citation. Some of the citation are after full stop and others are before full stop (newborn.7 vs visits 8.)

4. This part needs extensive grammar and citation editing and conceptual maturation of the problem.

Method

1. Your study was conducted from October 23 to December 17, 2020. It was almost five years ago. Why are you late to submit the article for publication? Data five years ago is not reliable for publication. What are the best practice standards for maintaining research records after completion for publication?

2. The sample size determination is good. But how did you come up with the design effect of 2? How can it be 2? Why not other numbers? You have to show.

3. You determined the sample size of an estimated 686 respondents during sample size calculation, but the study was conducted among 562 pregnant women. Why does this disparity occur (686 -562 = 124)?

4. Sampling procedure not clear, it is for interventional study or for your big project not for baseline study. Please clarify it.

5. Figure 1 is a trail flow diagram. Why is it important in a baseline study?

6. How do you measure institutional delivery services in this baseline study? Your study participants were pregnant mothers. Are you asking previous delivery service utilization?

7. Please clearly operationalize your outcome variable maternal health care services (antenatal care and institutional delivery services utilization).

Result

1. What is the importance of a detailed description of knowledge on obstetric danger signs among pregnant women? You can compute it and use it as one predictor variable for the outcome variable.

2. Why do you omit government employee from the category of both wife and husband occupation?

3. How do you compute the wealth index? Have you done PCA for it?

Discussion

1. You have to modify introductory paragraph for discussion

2 There is redundancy on at the beginning of discussion

3: Why not include the value of the predictors during discussion? Instead of saying, ‘Women with private employee husbands were more likely utilize antenatal care services than farmer husbands.’ You can write as women with private employee husbands were two times more likely utilize antenatal care services than farmer husbands

4 In general, the discussion part is somewhat good but needs few modification.

5 Can a researcher be an author and acknowledged at the same time? Please differentiate acknowledgement versus authorship. In your study, the co-author is also acknowledged. Why?

6 Please include URL for some references like R1, R2 and R3

6. PLOS authors have the option to publish the peer review history of their article (what does this mean? ). If published, this will include your full peer review and any attached files.

**Do you want your identity to be public for this peer review?** For information about this choice, including consent withdrawal, please see our Privacy Policy .

Reviewer #1: **Yes: ** Simachew Animen Bante

Reviewer #2: No

---

## [Author Response · Author response to Decision Letter 1]

19 Feb 2025

Editor and reviewers comments has been responded point by point response. please refer to response to reviewers.

---

## [Decision Letter · Decision Letter 1]

10 Mar 2025

PONE-D-24-46689R1Maternal Health Care Services Utilization Among Pregnant Women in Kersa District, Jimma Zone, Southwest, Ethiopia.PLOS ONE

Dear Dr. Abebe,

Thank you for submitting your manuscript to PLOS ONE. After careful consideration, we feel that it has merit but does not fully meet PLOS ONE’s publication criteria as it currently stands. Therefore, we invite you to submit a revised version of the manuscript that addresses the points raised during the review process.

Please submit your revised manuscript by Apr 24 2025 11:59PM. If you will need more time than this to complete your revisions, please reply to this message or contact the journal office at plosone@plos.org . Please include the following items when submitting your revised manuscript:

We look forward to receiving your revised manuscript.

Kind regards,

Kahsu Gebrekidan, Ph.D.

Academic Editor

PLOS ONE

Journal Requirements:

Reviewers' comments:

Reviewer's Responses to Questions

**Comments to the Author**

1. If the authors have adequately addressed your comments raised in a previous round of review and you feel that this manuscript is now acceptable for publication, you may indicate that here to bypass the “Comments to the Author” section, enter your conflict of interest statement in the “Confidential to Editor” section, and submit your "Accept" recommendation.

Reviewer #2: All comments have been addressed

Reviewer #3: All comments have been addressed

2. Is the manuscript technically sound, and do the data support the conclusions?

Reviewer #2: Yes

Reviewer #3: Yes

3. Has the statistical analysis been performed appropriately and rigorously? 

Reviewer #2: Yes

Reviewer #3: Yes

4. Have the authors made all data underlying the findings in their manuscript fully available?

Reviewer #2: Yes

Reviewer #3: Yes

5. Is the manuscript presented in an intelligible fashion and written in standard English?

Reviewer #2: Yes

Reviewer #3: Yes

6. Review Comments to the Author

Reviewer #2: (No Response)

Reviewer #3: Thank you for inviting me to review this manuscript. Please find my detailed comments below:

General Comment:

The clean copy of the manuscript does not appear to have been revised in accordance with the reviewers’ comments.

Introduction:

1. Line 109: When reporting statistics about the use of skilled birth attendance by urban versus rural women, it is essential to include the year of the investigation for context and accuracy.

2. Please provide information on the national guidelines for antenatal care (ANC) in Ethiopia. For example:

o At what gestational age do women typically begin ANC?

o How many ANC visits are recommended?

o Who is responsible for providing ANC services?

Methods:

1. Please clarify why women with a gestational age of 12–24 weeks were specifically recruited for this study.

2. Line 170: Since this is a cross-sectional study, randomization was not applicable. Instead, random selection was used. Please revise the text to reflect this distinction clearly.

3. I recommend that the authors clearly separate the description of this cross-sectional study from any interventional study. The current writing style may confuse readers about the study design.

Results:

1. Table 1: Please include the units of measurement for each variable to enhance clarity.

2. Table 2: The term “experienced pregnancy complications” is unclear. Please define what specific complications were included in this category.

3. Regarding “severe bleeding,” was this referring to postpartum hemorrhage (PPH)? If so, please specify.

4. Table 3: For “Home delivery,” please indicate whether a health provider (e.g., a midwife) was present during the delivery.

5. Table 4: Please specify the factors for which the adjusted odds ratios (ORs) were calculated.

Discussion:

1. One of the authors’ recommendations is to “increase household income.” This recommendation falls outside the scope of the Ministry of Health’s decision-making authority and would require significant government investment. Please consider revising this recommendation to focus on actionable steps within the health sector’s capacity.

7. PLOS authors have the option to publish the peer review history of their article (what does this mean? ). If published, this will include your full peer review and any attached files.

**Do you want your identity to be public for this peer review?** For information about this choice, including consent withdrawal, please see our Privacy Policy .

Reviewer #2: No

Reviewer #3: **Yes: ** Prof Parvin Abedi

---

## [Author Response · Author response to Decision Letter 2]

27 Mar 2025

Manuscript title: Maternal health care services utilization and associated factors among pregnant women kersa district, Jimma zone, southwest Ethiopia.

Dear Editor of the Plos One Journal, the revised manuscript of the above title has been submitted to your journal after carefully addressing the comments and suggestions provided.

sincerely,

Tsedach Alemu Abebe

Corresponding author

---

## [Decision Letter · Decision Letter 2]

8 Apr 2025

PONE-D-24-46689R2Maternal health care services utilization and associated factors among pregnant women in Kersa district, Jimma zone, Southwest Ethiopia.PLOS ONE

Dear Dr. Abebe,

Thank you for submitting your manuscript to PLOS ONE. After careful consideration, we feel that it has merit but does not fully meet PLOS ONE’s publication criteria as it currently stands. Therefore, we invite you to submit a revised version of the manuscript that addresses the points raised during the review process.

We look forward to receiving your revised manuscript.

Kind regards,

Kahsu Gebrekidan, Ph.D.

Academic Editor

PLOS ONE

Journal Requirements:

Reviewers' comments:

Reviewer's Responses to Questions

**Comments to the Author**

1. If the authors have adequately addressed your comments raised in a previous round of review and you feel that this manuscript is now acceptable for publication, you may indicate that here to bypass the “Comments to the Author” section, enter your conflict of interest statement in the “Confidential to Editor” section, and submit your "Accept" recommendation.

Reviewer #3: All comments have been addressed

2. Is the manuscript technically sound, and do the data support the conclusions?

Reviewer #3: Yes

3. Has the statistical analysis been performed appropriately and rigorously? 

Reviewer #3: Yes

4. Have the authors made all data underlying the findings in their manuscript fully available?

Reviewer #3: Yes

5. Is the manuscript presented in an intelligible fashion and written in standard English?

Reviewer #3: Yes

6. Review Comments to the Author

Reviewer #3: Thanks to the authors as they responded my previous comments comprehensively. However, the responses provided by the article's authors to each section of the introduction, methodology, and results were not incorporated into the paper and remained unavailable for review."

7. PLOS authors have the option to publish the peer review history of their article (what does this mean? ). If published, this will include your full peer review and any attached files.

**Do you want your identity to be public for this peer review?** For information about this choice, including consent withdrawal, please see our Privacy Policy .

Reviewer #3: **Yes: ** Prof Parvin Abedi

---

## [Author Response · Author response to Decision Letter 3]

9 Apr 2025

required documents have been uploaded.

---

## [Editor Report · Decision Letter 3]

17 Apr 2025

Maternal health care services utilization and associated factors among pregnant women in Kersa district, Jimma zone, Southwest Ethiopia.

PONE-D-24-46689R3

Dear Dr. Tsedach,

We’re pleased to inform you that your manuscript has been judged scientifically suitable for publication and will be formally accepted for publication once it meets all outstanding technical requirements.

Kind regards,

Kahsu Gebrekidan, Ph.D.

Academic Editor

PLOS ONE
---

## [Editor Report · Acceptance letter]

PONE-D-24-46689R3

PLOS ONE

Dear Dr. Abebe,

I'm pleased to inform you that your manuscript has been deemed suitable for publication in PLOS ONE. Congratulations! Your manuscript is now being handed over to our production team.

Kind regards,

on behalf of

Dr. Kahsu Gebrekidan

Academic Editor

PLOS ONE